# ADAMTS13, von Willebrand Factor, Platelet Microparticles, Factor VIII, and Impact of Somatic Mutations in the Pathogenesis of Splanchnic Vein Thrombosis Associated with BCR-ABL-Negative Myeloproliferative Neoplasms

**DOI:** 10.3390/life14040486

**Published:** 2024-04-09

**Authors:** Roberto Castelli, Alessandra Berzuini, Roberto Manetti, Alessandro Palmerio Delitala, Dante Castro, Giuseppe Sanna, Marta Chiara Sircana, Nicia Isabella Profili, Arianna Bartoli, Leyla La Cava, Giorgio Lambertenghi Deliliers, Mattia Donadoni, Antonio Gidaro

**Affiliations:** 1Department of Medicine, Surgery and Pharmacy, University of Sassari, 07100 Sassari, Italy; rmanetti@uniss.it (R.M.); d.castro@studenti.uniss.it (D.C.); g.sanna7@studenti.uniss.it (G.S.); m.sircana4@phd.uniss.it (M.C.S.); nicia.isa.profili@gmail.com (N.I.P.); 2Independent Professional Hematologist; aleberzuini@gmail.com; 3Department of Biomedical and Clinical Sciences Luigi Sacco, Luigi Sacco Hospital, University of Milan, 20157 Milan, Italy; arianna.bartoli@unimi.it (A.B.); leyla.lacava@unimi.it (L.L.C.); donadoni.mattia@asst-fbf-sacco.it (M.D.); 4Fondazione Matarelli, 20122 Milan, Italy; giorgio.lambertenghi@unimi.it

**Keywords:** thrombosis, microparticles, ADAMTS13, von Willebrand factor (VWF), factor VIII C (FVIII:C), splanchnic vein thrombosis, myeloproliferative neoplasms, somatic mutations, polycythemia vera, essential thrombocythemia, idiopathic myelofibrosis

## Abstract

Background: Myeloproliferative neoplasms (MPNs) are often associated with splanchnic vein thrombosis (SVT). Not all the factors involved in the thrombotic tendency are currently known. Objectives: This study aims to evaluate a possible association between ADAMTS13, von Willebrand factor (VWF), platelet microvesicles (MV), and factor VIII activity (FVIII:C) with thrombotic events in MPN patients. Materials and methods: In total, 36 consecutive MPN patients with SVT were enrolled. The MPNs were diagnosed based on clinical characteristics and one or more gene mutations among JAK-2, CALR, and MPL. As controls, 50 randomly selected patients with MPN without thrombosis, 50 patients with deep vein thrombosis without MPNs, and 50 healthy blood donors were evaluated. Complete blood count, ADAMTS13, VWF, MV, and FVIII:C in plasma were measured in all the subjects. Results: The JAK-2 mutation was found in 94% of the patients with SVT, but none were triple-negative for genetic mutations (JAK2 V617F, CALR, MPL, and exon 12). Compared to the normal subjects, in all the MPN patients (with or without SVT), the levels of ADAMTS13 were found to be significantly lower (*p* < 0.001) and the MV concentrations were significantly higher (*p* < 0.001). Among the MPN patients, the VWF and FVIII:C levels were significantly higher in the patients with SVT than those without thrombosis (*p* = 0.007 and *p* = 0.04, respectively). Splenomegaly was present in 78% of MPN patients with SVT and in 30% of those without SVT (*p* < 0.001). The ADAMTS13/VWF ratio was reduced in all the patients, but not in the healthy blood donors (*p* < 0.001). Conclusions: The significant increase in circulating MV, VWF, and FVIII:C in the MPN patients and in the patients with thrombosis supports the role of endothelium damage in promoting thrombotic events. In particular, a significant increase in VWF and FVIII:C levels was found in the MPN patients with SVT.

## 1. Introduction

Myeloproliferative neoplasms (MPNs) are clonal diseases of bone-marrow stem cells, leading to any combination of increased red blood cells (RBC), white blood cells (WBC), and platelets (PLTs) in peripheral blood [1]. Polycythemia vera (PV), essential thrombocythemia (ET), and idiopathic myelofibrosis (IMF) are the main clinical manifestations. The global incidence of MPNs is 4/100,000 patients/year. Focusing on each singular manifestation, PV incidence is 2/100,000 patients/year, ET is 1.3/100,000, and IMF is 0.5/10,000 patients/year [2].

The symptoms of MPNs are heterogeneous. The most commonly represented are fatigue (81 percent), pruritus (52 percent), night sweats (49 percent), bone pain (44 percent), fever (14 percent), and weight loss (13 percent). Fatigue was the most frequently reported symptom among each MPNs as it was present in 85%, 72%, and 84% of people with PV, ET, and PMF, respectively [3]. Approximately one-third of the participants needed assistance with daily living activities, and 11% reported medical disability related to MPNs [3].

Patients with PV and ET both exhibit mild symptoms because of vasomotor alteration, including headache, lightheadedness, syncope, atypical chest pain, acral paresthesia, livedo reticularis, erythromelalgia (burning pain in the hands or feet associated with erythema and warmth), and transient visual disturbances (e.g., amaurosis fugax, scintillating scotomata, ophthalmic migraine) [3].

Although the exact causes of vasomotor symptoms are unknown, studies have linked erythromelalgia with thromboxane-dependent platelet activation and consequent arterial microvascular thrombosis. These events are usually more annoying than dangerous, and low or standard doses of aspirin are frequently used to treat them [4].

The diagnostic criteria are based on the presence of major molecular hematopoietic drivers: the most frequent are the V617F mutation in the JAK2 gene, mutations in exon 10 of the MPL gene (mainly involving codon W515), which are present in 90% of patients with PV, and JAK2 mutations in exon 12 [5].

Recently, new frameshift mutations in exon 9 of the calreticulin (CALR) are found in around 70% of ET or PMF patients who do not have JAK2 or MPL mutations, but rarely in PV patients. Several studies report CALR mutations in approximately 20% to 25% of patients with ET and IMF but not in PV [6]. The MPL mutation is present in 4–5% of JAK2-negative patients, whereas IMF is a complex condition characterized by JAK2 V617F mutation and CALR-gene mutation, in which myeloproliferation often coexists with anemia or thrombocytopenia [5,6,7,8,9].

There are still no known explanations for the broad range of phenotypic variations linked to a single gene mutation. However, it has been suggested that gene-dosage effects (allele burden), other genetic events, and genetic instability may play a role.

Blood clot formation, excessive bleeding, and the development of acute myeloid leukemia (AML) or a fibrotic phase of the disease are some of the primary problems linked to MPNs [10].

Together, MPNs display a propensity towards clonal evolution and disease progression to AML, MDS, or a fibrotic phase. The likelihood of transforming into AML varies across the subgroups, with CML having the highest risk (over 90 percent without effective therapy) and ET having the lowest risk (less than 5 percent). Patients with PV face a likelihood of approximately 10 and 25 percent of progressing to a myelofibrotic stage after 10 and 25 years of follow-up, respectively [11].

The literature reports that the incidence of thrombotic events is around 11–25% in ET, 12–39% in PV, and about 9.5% in IMF [12].

According to the literature, the rates of arterial and venous thromboses are higher in patients with all categories of MPN, regardless of age or gender [12]. The incidence of thrombosis is higher soon after diagnosis. As Mora et al. demonstrated, the incidence of arterial and venous thromboses in MPNs patients during the first three months after diagnosis, compared with the control population, was three- and ten-fold higher, respectively. While the hazard ratios declined in subsequent years, they remained elevated throughout the five-year observation period [12].

The reduction in thrombosis risk could be explained by the reduction in blood cells after therapy starts. Recently, some evidence confirmed high values of red blood cells and platelets lead to thrombosis, but also neutrophils and monocytes [13]. Patients with MPNs have been found to have an additional element, known as neutrophil extracellular traps (NETs). These are networks of extracellular fibers and DNA-encompassing assemblies derived from neutrophils. They are essential in fighting infections and innate host-defense mechanisms, but they can also cause sterile inflammation [13].

Elaskalani et al. found that cell-free NETs obtained from LPS-stimulated neutrophils can promote platelet aggregation when incubated with human platelets [14]. In the study, it was also observed that the platelets’ surface expression of CD62 and phosphatidylserine (PS) increased, indicating that NETs activate intracellular signaling in platelets [14]. These results strongly suggest that neutrophil function plays a crucial role in thrombosis.

The assessment of thrombotic risk is a crucial aspect of management for patients with MPNs.

The literature suggests that using low-dose aspirin rather than cytoreductive therapy to initially treat low-risk MPNs. This approach is considered standard therapy for patients at low risk of disease because of the favorable prognosis and the potential adverse effects of cytoreductive agents. Once-initiated aspirin is also maintained in the higher-risk-score class [15].

Aspirin should not be used, or used carefully, in patients with clinically significant acquired von Willebrand syndrome or a very-low-risk score (Age ≤60, no JAK2 mutation, and no history of thrombosis) [16].

The literature suggests twice-daily low-dose aspirin rather than a single dose of low-dose aspirin, using no more than 100 mg daily, for patients with inadequate control of vasomotor symptoms [17].

It is essential to use a total daily aspirin dose ≤100 mg (e.g., aspirin 42 mg twice daily by mouth) to avoid the excessive bleeding that ET-associated platelet-function abnormalities may exacerbate [17].

Although there is no evidence to support divided dosing over a single daily dose for the management of vasomotor symptoms, some people experience better symptom control when taking their medication twice daily. Additionally, there is biochemical evidence that divided doses are more effective for suppressing platelet cyclooxygenase 1 (COX-1) activity (the target of action) [17].

Cytoreductive treatment (e.g., hydroxyurea) is the second line of therapy in high-risk MPNs [18]. Depending on the type of MPN, indications to start the treatment differ. In PV, cytoreductive treatment is recommended if phlebotomy does not keep the hematocrit (Hct) at less than 45 percent [18]. In high (history of thrombosis or age >60 with JAK2 V617F)- and intermediate (age >60, no JAK2 mutation, and no history of thrombosis)-risk ET, cytoreductive therapy is mandatory to obtain both symptom relief and reduction in cardiovascular risk [18]. According to current guidelines, JAK inhibitors like ruxolitinib or hydroxyurea are recommended when an IMF patient exhibits symptomatic splenomegaly without accompanying anemia [18].

However, thrombotic events still occur in 10% of patients receiving cytoreductive and aspirin treatment [12]. If venous thrombosis occurs, anticoagulation therapy should be started.

The SVT is a rare site of thrombosis (1%–10% among all types of MPN). The literature reports JAK2 V617F substitution in 32.7% of patients with SVT, although, in one-third of cases, SVT is incidentally detected during routine abdominal ultrasound performed for other reasons [16,19,20]. Furthermore, SVT may involve portal veins (the most common), mesenteric or splenic veins, and, less commonly, hepatic veins (Budd Chiari syndrome, BCS).

In our study, we aimed to evaluate the possible role of markers of endothelial damage, such as ADAMTS13, platelet-derived microvesicles (MVs), von Willebrand factor (VWF), and Factor VIII activity (FVIII:C), in the pathogenesis of SVT in MPN patients.

## 2. Materials and Methods

### 2.1. Study Population and SVT Diagnosis

We prospectively enrolled a consecutive series of MPN patients with SVT admitted to our hospital from 1 January 2021 until 31 December 2022. Thirty-six patients (25 females and 11 males, median age 50 years) fulfilled the diagnostic criteria for MPNs and the simultaneous presence of SVT, including the absence of acute inflammation (CRP < 10 mg/L in two measures after hospital admission), normal liver function, coagulation tests, and negative autoimmunity (Group A). For the control group, we analyzed 50 MPN patients, comparable in terms of age and sex, without SVT (Group B), 50 patients with deep vein thrombosis (DVT) without MPNs (Group C), and 50 healthy donors (Group D). Complete blood count, ADAMTS13, MV, VWF, FVIII:C, and AB0 typing were determined in all groups. Diagnosis of MPNs was based on hematological parameters and molecular analysis [7], and, in 20 cases, it was confirmed by morphological analysis of bone marrow specimens according to WHO morphological criteria. All the MPN patients underwent an abdominal ultrasound to detect a possible SVT and, thus, they were included in group A if positive or group B if negative. According to recognized criteria, asymptomatic SVT has never been seen in any subject [20,21]. The DVT diagnosis was based on the following criteria: (1) evidence of DVT of the lower limbs at compression ultrasonography; (2) exclusion of cancer, surgery, or immobilization; (3) regular coagulation tests; (4) absence of lupus anticoagulant, FV Leiden and prothrombin G 20210A mutations. The AB0 blood type, recognized for its possible role in thrombotic events, was also checked in all subjects.

The study was conducted according to the guidelines of the Declaration of Helsinki. All study participants gave written informed consent, and the local ethical committee approved the study.

### 2.2. Sample Collection and Storage

Blood samples were collected on the first day of the hospital stay (for groups A and C) before heparin started. All the blood samples were collected after 12 h of fasting by antecubital vein venipuncture, using sodium citrate 3.8% as an anticoagulant. Plasma was obtained by centrifuging samples at 2000× *g* for 20 min at room temperature, and then divided in small aliquots and stored at −80 °C until testing.

### 2.3. Measurements

Citrated blood specimens for ADAMTS13 and VWF Ag assays were centrifuged at 2000× *g* for 20 min at 20 °C and aliquots were stored at −70 °C. The VWF antigen was measured in citrated plasma by an “in-house” sandwich ELISA using two monoclonal antibodies directed against different VWF epitopes (11B6.18 and 7G10.8). The ADAMTS13 activity was assayed using collagen-binding enzyme immunoassay [22]. According to the literature, a commercial one-stage assay measured FVIII:C [23]. The AB0 typing was performed using genotyping techniques [24]. A correction factor on FVIII and VWF was applied according to the AB0 group, as reported in the literature (normally, 0-blood-group levels are around 20–30% lower than those of non-0-groups. To correct this bias, 0-group levels were increased by 30%) [25]. According to the literature, MV levels were measured immediately after blood sampling in platelet-free plasma [26].

Bone marrow specimens were stained with hematoxylin, eosin, Giemsa, and Gomori silver. Bone marrow fibrosis was graded based on the EUMNET consensus [27]. JAK2 V617F mutation was searched using allele-specific PCR using the protocol developed by Baxter et al. [28]. The MPL and CALR mutations were evaluated according to the literature [7].

### 2.4. Statistical Analysis

Statistical analysis was performed using SPSS (Statistical Package for Social Science-SPSS, Inc., Chicago, IL, USA, version 20). Normal distribution was evaluated using the Kolmogorov–Smirnov test. Qualitative data were expressed as numbers and percentages. Fisher’s exact tests were used for group comparison. Quantitative data were described as mean, standard deviation, median, and interquartile range. Statistical differences were investigated with analysis of variance (ANOVA or mixed-effect analysis), while specific differences were assessed with Holm–Bonferroni correction for multiple comparisons. A *p*-value < 0.05 was considered statistically significant.

## 3. Results

Table 1 describes the study populations’ demographics, clinical data, and mutational status.

Bone marrow evaluation was performed on specimens obtained from 20 patients (13 patients in the A group and 7 in the B group). In 55% of the Group A patients, hypercellular bone marrow was found, and 50% showed increased erythropoiesis and/or granulopoiesis. A loss of megakaryocytes was observed in 15% of the samples. Based on the hematological and morphological features, 36% of the Group A patients were classified as PV, 31% as ET, and 33% as IMF. In Group B, 38% of the patients were classified as PV, 30% as ET, and 32% as IMF. The JAK2 V617F mutation was found (with a 25% burden) in 94% of the Group A patients and 86% of the Group B patients (*p* = 0.5, not significant).

Patients negative for JAK2 V617F were evaluated for CALR and MPL mutations, which are mutually exclusive. In Group A, one patient showed CALR, and one showed MPL mutations. No patients were found to be triple-negative.

Splenomegaly, defined by a spleen diameter greater than 13 cm, was present in 78% of the Group A patients and 30% of the Group B patients (*p* < 0.001).

A previously unrecognized drive mutation (one JAK2 V617F, one CALR, and one MPL) was found in three Group C patients without any clinical features of MPN. These patients were excluded from the analysis and followed in a clinical program.

The ADAMTS13, MV, VWF, and FVIII:C levels among all the study populations are shown in Figure 1. The MVs were significantly higher in all the patients (Groups A, B, C) than in the controls, in Group D (*p* < 0.001) (Figure 1(2)). According to the univariate analysis, the ADAMTS13 levels were significantly lower in the MPN patients (*p* < 0.001), regardless of the presence or absence of SVT (Figure 1(1)).

In all the patients (Groups A, B and C), the VWF and FVIII:C levels were significantly increased in relation to the control, group D (*p* < 0.001) (Figure 1(3,4)). Among the MPN patients, higher levels of VWF and FVIII:C were found in Group A compared to Group B (*p* = 0.007 and *p* = 0.04). The VWF and FVIII:C were increased in the non-0-blood-group patients in relation to the 0 blood type (respectively 345 + 180 vs. 275 + 44 *p* = 0.047; 190 + 106 vs. 148 + 44 *p* = 0.048).

The ADAMTS13/VWF ratio was reduced in all the patients, except for the healthy blood donors (Figure 1(5)).

The absolute number of monocytes and neutrophils was higher in both the MPN groups (A and B) compared to the patients with thrombosis and the healthy volunteers (C and D) (Figure 1(6,7)).

Anticoagulation was started in all the patients from Groups A and C; the first group received heparin alone, while the second received heparin, and, after one week, direct-acting oral anticoagulants based on physician choice.

## 4. Discussion

Myeloproliferative neoplasms are clonal hemopoietic stem cell disorders characterized by a high incidence of thrombotic (either arterial or venous) and hemorrhagic complications. In addition to factors such as age >60 years and previous thrombotic events, JAK2 V617F mutation has been validated as a thrombotic risk factor because it promotes thrombus formation through the overexpression of P selectin, leading to platelet aggregation and fibrin deposition [29,30,31,32,33,34]. Thrombosis is a common occurrence in PV, ET, and IMF, often preceding other clinical manifestations. [27]. In our case series, presumably due to the low numbers, we did not observe any differences between the MPN subsets. On the other hand, our study confirms that MPN-SVT mostly afflicts women rather than men, as shown in the literature.

None of the patients had liver cirrhosis (the most frequently associated disease) or active COVID-19 infection. The Bcr/abl-negative MPN is a leading cause of SVT, accounting for 40% of BCS and 30% of extra-hepatic portal vein obstructions, while other MPN molecular markers, such as mutations in JAK2 exon 12, CALR, and MPL genes, are very rarely recognized as promoters of thrombotic events [20].

In our small cohort, SVT was represented by portal vein thrombosis in 76%, splenic vein thrombosis in 18%, and mesenteric vein thrombosis in 6%.

In 94% of the patients, SVT was associated with JAK-2 gene mutation and, in two cases, with CALR and MPL W515K, respectively. It is worth noting that the JAK2 V617F mutation is a critical factor in MPN, which is associated with SVT. However, around 14–20% of SVT patients with MPN do not have this mutation [35]. Regarding the clinical risk, patients with JAK2-unmutated MPN-SVT do not statistically differ from those with JAK2-mutated MPN-SVT. This leads to the question of whether other mutations contribute to the prothrombotic state. Various mutations with an increased incidence in MPNs have been identified, and researchers have only recently begun to explore their role in SVT. Furthermore, CALR mutations are rare in patients with SVTs, with a pooled proportion of only 0–4.88%. However, in JAK2-V617F-negative MPN-SVT patients, the pooled proportion of CALR mutations was found to be 15.16% in a recent meta-analysis [36].

Although some contributing mechanisms for the formation of blood clots have been partially specified in recent years, many points remain unexplained. These mechanisms include the overproduction of blood cells, platelet abnormalities, increased endothelial activation, or neo-angiogenesis [37,38].

In our cohort, splenomegaly was more frequent in the MPN-SVT patients than in the MPN patients. This situation can lead to the sequestration of blood cells and hemodilution, effectively concealing the typical clinical features of MPNs, such as elevated peripheral blood cell counts.

In this study, we evaluated the clinical, hematological, histopathological, and molecular features of MPN patients with or without thrombosis, focusing on possible associations with thrombotic events and comparing them with non-MPN subjects. In particular, we concentrated on the ADAMTS13, MV, VWF, and FVIII:C levels in different subgroups to exclude coexistent coagulation abnormalities or inflammation.

The well-recognized influence of the non-0 blood group on thrombus formation (due to increased levels of circulating FVIII) was confirmed in our cohort, in which all the SVT patients belonged to the non-0 blood group (with the A blood group accounting for 75% and the B blood group accounting for 25%) [25].

We also explored the neutrophil and monocyte counts to evaluate a possible association with SVT. Neutrophils are known to play a central role in generating the inflammatory response and the activation of the blood coagulation system through the release of proteolytic enzymes, reactive oxygen species, and the increased expression of CD11b, which activates or damages platelets, endothelial cells, and coagulation proteins [39]. Neutrophil extracellular traps (NETs), released by neutrophils, may also be involved in thrombus formation [40]. The monocytes and neutrophils were higher in the MPN patients compared to Groups C and D; nonetheless, no statistical difference was found between Groups A and B.

The neutrophil-to-lymphocyte ratio is an established thrombosis risk factor for MPN patients [41], and it also affects the prognosis [42]. For this reason, in this study, we decided not to repeat the evaluation.

Recently, a JAK2 V617F mutation was also detected in the monocyte endothelial cells of Philadelphia-negative myeloproliferative diseases [33], adding a novel promoting factor in thrombosis risk.

Platelet MVs, recognized as thrombosis promoters, are generated by the outward budding and shedding of the plasma membrane with diameters ranging from 50 to 1000 nm. These MVs are present in the blood of healthy individuals, but their numbers and cellular sources are related to age and plasma levels of factors controlling inflammation, coagulation, and fibrinolysis [43]. Compared with the healthy controls, all the patients displayed higher levels of MVs (*p* < 0.001), but no significant differences were found among the A, B, and C groups. The MV measurement shows some limitations related to possible interferences [44]. In our study, we tried to reduce these biases through a strict blood sample collection method and by excluding all the patients with high inflammation, but we cannot exclude the possibility that some of them may have influenced our results. The lower levels of ADAMTS13 found in the A, B, and C subgroups are likely due to the elevation of VWF, FVIII:C, and MVs [45,46]. This hypothesis is supported by the significantly higher levels of VWF and FVIII:C found in the Group A patients; concerning the Group B patients, VWF is released by damaged endothelial cells, increases blood viscosity, boosts inflammation with cytokine release, and promotes microthromboses. In MPNs, an increase in circulating endothelial cells has been described as a result of endothelial damage [47]. The ADAMTS13/VWF ratio is considered a sensitive index to evaluate blood flow in liver-related splanchnic vein thrombosis. We found a low ratio in all the patients (Groups A, B, C); nonetheless, we could not identify prognostic factors of thrombosis in the MPN patients in our series [48,49,50].

As a limit of our work, we did not evaluate the numerous non-driver somatic mutations affecting the genes responsible for various cellular processes identified in MPN (DNA methylation: TET2, DNMT3A, IDH1-IDH2; chromatin spliceosome: ASXL1, EZH2, SF3B1, SRSF2, U2AF1, ZRSF2; and others, such as TP53) [35]. This choice was defined because many of these mutations have been found in otherwise healthy individuals, and their prevalence increases with age. Therefore, we do not know their role in the MPN pathological process.

## 5. Conclusions

Our study features several limitations, since it is a prospective observation of a limited case series from a single center. It also lacks the observation of patients with SVT without MPNs.

Nonetheless, this study represents a preliminary approach to understanding the pathogenesis of splanchnic vein thrombosis in MPN patients. For this reason, these incomplete data suggest the need for new prospective and multicentric studies.

## Figures and Tables

**Figure 1 life-14-00486-f001:**
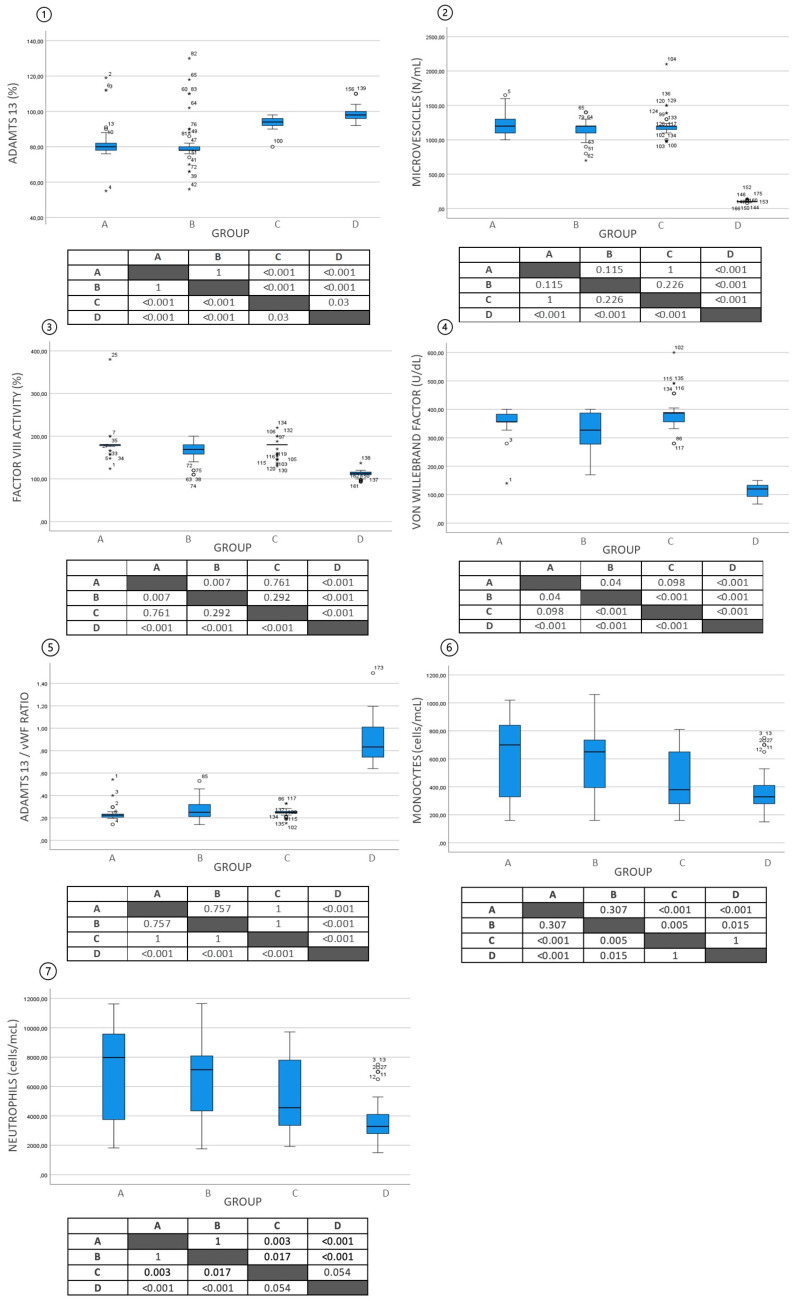
ADAMTS13 (**1**), microvesicles (**2**), Factor VIII (**3**), von Willebrand factor (VWF) (**4**), ADAMTS13/VWF ratio (**5**), monocytes (**6**), neutrophils (**7**) in the four groups. In the tables: *p* values. (*) = outliers.

**Table 1 life-14-00486-t001:** Clinical characteristics of the study population. Group A: patients with MPN and SVT; Group B: patients with MPN without thrombosis; Group C: patients with thrombosis without MPN; Group D: healthy blood donors.

	Group A N = 36	Group B N = 50	Group C N = 50	Group D N = 50
Age (IQR)—yr	50 (36–68)	52 (37–70)	64 (28–75)	42 (20–60)
Male sex (%)—no.	11 (30)	20 (40)	10 (20)	20 (40)
AB0 blood group (%)—no.				
A	27 (75)	27 (54)	25 (50)	27 (54)
B	9 (25)	10 (20)	10 (20)	13 (25)
AB	0	0	2 (4)	0
O	0	13 (26)	13 (26)	0
MPN type (%)—no.				
PV	13 (36.1)	19 (38)	0	0
ET	11 (30.5)	15 (30)	0	0
IMF	12 (33.3)	16 (32)	0	0
Hemoglobin (SD)—g/dL	12 ± 6.9)	11.8 ± 5	14.6 ± 6.4	15.3 ± 4.6
WBC count (SD)—×10^9^/L	9.5 ± 10.9	9.8 ± 11.6	9 ± 9.5	5.6 ± 4.2
Absolute neutrophils (SD)—×10^9^/L	6.9 ± 0.48	6.6 ± 0.32	5.1 ± 0.3	3.7 ± 0.3
Absolute monocyte (SD)—×10^9^/L	0.6 ± 0.28	0.62 ± 0.22	0.47 ± 0.19	0.37 ± 0.18
PLT count (SD)—×10^9^/L	445 ± 621	454 ± 641	296 ± 336	261 ± 267
LDH (SD)—IU/L	962 ± 1451	555 ± 733	212 ± 228	204 ± 237
Splanchnic vein thrombosis subtype (%)—no.				
Portal vein thrombosis	27 (76)	0	0	0
Splenic vein thrombosis	6 (18)	0	0	0
Mesenteric vein thrombosis	2 (6)	0	0	0
Budd Chiari syndrome	1 (3)	0	0	0
Splenomegaly (diameter >13 cm) no. (%)	28 (78)	15 (30)	0	0
Acetylsalicylic acid no. (%)	27 (75)	37 (74)	14 (28)	5 (10)
Cytoreductive therapy (hydroxyurea) no. (%)	24 (67)	34 (68)	0	0
Molecular analysis (%)—no.				
JAK2 V617F	34 (94.4)	43 (86)	1 (2)	0
CALR	1 (2.7)	6 (12)	1 (2)	0
MPL W515K	1 (2.7)	1 (2)	1 (2)	0
Triple-negative	0	0	47 (94)	50 (100)
JAK2 allele burden (IQR)—% of activity	25 (4.8–97)	30 (5–95)	5	0
ADAMTS13 (SD)—% of activity	82.3 ± 11.5	82.1 ± 12.6	93.5 ± 3.0	98.3 ± 3.9
Von Willebrand factor VWF (SD)—U/dL	357.3 ± 43.9	318.8 ± 70.7	383.9 ± 49.9	116.5 ± 20.8
FVIII:C (SD)—IU/dL	181.8 ± 30.9	165.5 ± 23.3	174.2 ± 16.9	114.7 ± 8.3
ADAMTS13/VWF—ratio	0.273 ± 0.067	0.275 ± 0.09	0.247 ± 0.029	0.874 ± 0.018
Microvesicles level (SD)—N/mL	1231 ± 159.9	1162 ± 134.4	1217 ± 168.8	105 ± 15.9

IQR: interquartile range; SD: standard deviation; yr: years; MPN: myeloproliferative neoplasm; SVT: splanchnic vein thrombosis; PV: polycythemia vera; ET: essential thrombocythemia; IMF: idiopathic myelofibrosis; WBC: white blood count; PLT: platelets.

## Data Availability

The study data will be available to the corresponding author upon request.

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
