# Peer review of "ADAMTS13, von Willebrand Factor, Platelet Microparticles, Factor VIII, and Impact of Somatic Mutations in the Pathogenesis of Splanchnic Vein Thrombosis Associated with BCR-ABL-Negative Myeloproliferative Neoplasms"

_life, 2024, doi:10.3390/life14040486_

Round 1

Reviewer 1 Report

Comments and Suggestions for Authors

This is an interesting and mostly well-written article exploring a number of variables pertinent to the coagulation cascade, such as ADAMTS13 levels, vWF and factor VIII levels, platelet microparticles, etc. in 36 patients with MPN and splanchnic vein thrombosis, who are compared to 50 patients with MPN but no thrombosis, 50 with DVT but no MPN and 50 healthy blood donors. It is a retrospective study. Unsurprisingly, the researchers find that patients with MPNs have lower levels of ADAMTS13 and higher concentrations of platelet microparticles and of them, that those with SVT had higher levels of vWF and Factor VIII. Also expectedly, JAK2 mutations were clearly over-represented among the patients with SVT, and there was also a correlation between splenomegaly and SVT. A few points below for consideration:

1.      There are some typos that need to be corrected. For example, the statement that only 4-5% of patients with MPNs that lack JAK2 or MPL mutations harbor CALR mutations is obviously wrong. CALR mutations are found in the majority of JAK2-/MPL-WT patients. Ruxolitinib is badly mis-spelt.

2.      I read with great interest the section that claims that HU is the agent of choice ahead of JAK inhibitors for symptomatic splenomegaly in MF. The authors state that this is what the guidelines say. Which guidelines? JAK inhibitors are far superior to HU for the relief of splenomegaly and symptoms in patients with MF, and have been shown to be so in RCTs. Furthermore, they are generally very well-tolerated over prolonged periods, at least ruxolitinib is. The authors should delete this misleading and inaccurate section of the text, which also is not relevant to this particular subject.

3.      The authors examined neutrophil and monocyte counts and their relations to thrombosis. What about lymphocyte counts? The latter is particularly relevant given the demonstration that a high neutrophil to lymphocyte ratio correlates with venous thrombosis risk in PV.

4.      Although the article is not written by native speakers of English, it is, to the authors’ credit, generally easy to follow. However, it does need to be reviewed by a native speaker of English or someone highly proficient in the language. For example, the statement about “women who respect men” makes no sense and needs to be reworded.

Comments on the Quality of English Language

See last point above.

Author Response

Dear Editor,

Enclosed is our revised version of the manuscript entitled “ADAMTS13, von Willebrand Factor, Platelet MIcroparticles, Factor VIII, and Impact of Somatic Mutations in the Pathogenesis of Splanchnic Vein Thrombosis Associated with Bcr-Abl Negative Myeloproliferative Neoplasms” We want to thank the Editor and the reviewers for their constructive criticism, which substantially improved our paper. We hope that the paper will be suitable for publication. We have made a point-by-point answer to the referees’ comments below.

Answers to Reviewer 1

Reviewer 1 comment: I have the following specific suggestions/comments:

  1. Not sure why the title shows strange text, and why not corrected by authors before submission: MI-Croparticles, Nega-Tive

Response to Reviewer 1 comment:  We thank the reviewer for his/her correction. In the revised version, we corrected the title; the problem was an automatic slice of the word due to a line change.

Reviewer 1 comment: 2. The current recommended abbreviation for vWF is 'VWF; please change throughout. The recommended abbreviation for FVIII C is 'FVIII:C'; please change throughout.

Response to Reviewer 1 comment:  We thank the reviewer for his/her correction. In the revised version, we change the abbreviation as recommended.

Reviewer 1 comment: 3. Introduction; entire page 2 and page 3 contains many paragraphs, and statements of fact, but hardly any references; please include a reference for each statement of fact. Occasionally, authors state 'Several studies report.." and then list only a single reference. Many times they state 'In literature,' or 'Literature suggests', but fail to provide a reference.

Response to Reviewer 1 comment: We thank the reviewer for his/her correction. In the revised version, we added 7 references to support the introduction section on pages 2 and 3.

Reviewer 1 comment: 4. Page 4: "The study protocol was by the declaration of Helsinki." ?

Response to Reviewer 1 comment:  We thank the reviewer for his/her correction. In the revised version, we write the correct sentence: “The study was conducted according to the guidelines of the Declaration of Helsinki”.

Reviewer 1 comment: 5. Page 4: "centrifuging samples at 2000×g for 20 min at room temperature, then divided in small aliquots and stored at −80 °C until testing." Including for microparticles? Also for lupus anticoagulant?

Response to Reviewer 1 comment:  We thank the reviewer for his/her question. Both microparticles and lupus anticoagulant processes were different. We didn’t report the exact method for microparticles described in our previous paper (quote number 17) in the paper because the MDPI automatic check of plagiarism does not consent to the repetition.

Reviewer 1 comment: 6.  "ADAMTS-13 activity was assayed using collagen-binding enzyme immunoassay." may be misunderstood, since these assays look for the ability of plasma to cleave VWF, with VWF:CB as a detection system. Does Ref 13 as cited really describe the assay they used?

Response to Reviewer 1 comment:  We thank the reviewer for his/her question. We didn’t report the exact method for ADAMTS-13 activity described in our previous paper (quote number 13) in the paper because the MDPI automatic check of plagiarism does not consent to the repetition.

Reviewer 1 comment: 7. "According to the literature, a commercial chromogenic substrate assay measured FVIII C [14]." Does ref 14 as cited really describe the assay they used? Most labs use a one stage assay, not a chromogenic assay. Did you authors actually measure FVIII:C, or get their local hemostasis lab to measure FVIII:C?

Response to Reviewer 1 comment: We thank the reviewer for his/her correction. As the reviewer correctly indicated, the measure used a one-stage assay.

Reviewer 1 comment: 8. "A correction factor on FVIII and vWF was applied according to the AB0 group as reported in the literature" ? what correction factor? not mentioned in results.

Response to Reviewer 1 comment: We thank the reviewer for his/her question. According to the literature, the revised version specifies that blood group 0 measurement levels were increased by 30%.

Reviewer 1 comment: 9. Table 1: Not labelled as such, and no footer describing table abbreviations. Group D FVIII:C listed as 11.47±8.3; authors may mean 114.7±8.3? Check other data entries for errors.

Response to Reviewer 1 comment:  We thank the reviewer for his/her correction. We revised the data, corrected the error, and put the footer describing table abbreviations in the revised version.

Reviewer 1 comment: 10. Ref list: Ref 28 and 29 are one ref split into 2; authors should check/correct in text citations from ref 28.

Response to Reviewer 1 comment:  We thank the reviewer for his/her correction. We unify the 2 quoted in the revised version and adjust the reference list.

Answers to Reviewer 2

Reviewer 2 comment: - Please pay attention to typos in the title

Response to Reviewer 2 comment:  We thank the reviewer for his/her correction. In the revised version, we corrected the title; the problem was an automatic slice of the word due to a line change.

Reviewer 2 comment: - Page 4, line 149: please provide more evidence concerning absence of acute inflammation.

Response to Reviewer 2 comment:  We thank the reviewer for his/her suggestion. In the revised version, we add this sentence ” (PCR< 10 mg/L in two measures after hospital admission)”

Reviewer 2 comment: - Introduction: as this manuscript is more about diagnosis and markers of a pathological condition, talking about symptoms in three paragraphs (page 2, paragraph 2,3,4) is not suitable.

Response to Reviewer 2 comment:  We thank the reviewer for his/her suggestion. We decided not to cut the introduction because the majority of these symptoms have a vasomotor alteration as the origin, and our work supports the role of endothelium damage in promoting thrombotic events and these symptoms.

Reviewer 2 comment: - The same issue with treatment of this kind of patients in almost all page 3. Instead, they could provide more information about neutrophil and monocyte roles in this context.

Response to Reviewer 2 comment: We thank the reviewer for his/her correction. Based on two reviewers' indications, we decided to cut the introduction about splenomegaly.

On the other hand, a brief introduction about neutrophil and monocyte roles was added to the introduction as requested by the reviewer.

Reviewer 2 comment: - Page 9, line 252: please rephrase your sentence.

Response to Reviewer 2 comment: We thank the reviewer for his/her correction. We rephrased the sentence in the revised version.

Reviewer 2 comment: - Page 9, line 275-278: please rephrase your sentence.

Response to Reviewer 2 comment: We thank the reviewer for his/her correction. We rephrased the sentence in the revised version.

Reviewer 2 comment: - Some proteins or antibodies interfere with the tests the authors had been performed. Do the authors have information about fibrinogen concentration, rheuma factor, and lupus anticoagulant for these patients?

Response to Reviewer 2 comment:  We thank the reviewer for his/her question. The revised version specifies “ normal liver function, coagulation tests, and negative autoimmunity.” In the first version, these inclusion criteria were specified only for group D.

Answers to Reviewer 3

Reviewer 3 comment:

This is an interesting and mostly well-written article exploring a number of variables pertinent to the coagulation cascade, such as ADAMTS13 levels, vWF and factor VIII levels, platelet microparticles, etc. in 36 patients with MPN and splanchnic vein thrombosis, who are compared to 50 patients with MPN but no thrombosis, 50 with DVT but no MPN and 50 healthy blood donors. It is a retrospective study. Unsurprisingly, the researchers find that patients with MPNs have lower levels of ADAMTS13 and higher concentrations of platelet microparticles and of them, that those with SVT had higher levels of vWF and Factor VIII. Also expectedly, JAK2 mutations were clearly over-represented among the patients with SVT, and there was also a correlation between splenomegaly and SVT. A few points below for consideration:

Response to Reviewer 3 comment:  We thank the reviewer for his/her positive comment.

Reviewer 3 comment:

  1. There are some typos that need to be corrected. For example, the statement that only 4-5% of patients with MPNs that lack JAK2 or MPL mutations harbor CALR mutations is obviously wrong. CALR mutations are found in the majority of JAK2-/MPL-WT patients. Ruxolitinib is badly mis-spelt.

Response to Reviewer 3 comment:  We thank the reviewer for his/her correction. In the revised version, we corrected the sentence. Moreover, we precise the mis-spelt.

Reviewer 3 comment:

  1. I read with great interest the section that claims that HU is the agent of choice ahead of JAK inhibitors for symptomatic splenomegaly in MF. The authors state that this is what the guidelines say. Which guidelines? JAK inhibitors are far superior to HU for the relief of splenomegaly and symptoms in patients with MF, and have been shown to be so in RCTs. Furthermore, they are generally very well-tolerated over prolonged periods, at least ruxolitinib is. The authors should delete this misleading and inaccurate section of the text, which also is not relevant to this particular subject.

Response to Reviewer 3 comment:  We thank the reviewer for his/her correction. In the revised version, we deleted the sentence as the reviewers suggested.

Reviewer 3 comment:

  1. The authors examined neutrophil and monocyte counts and their relations to thrombosis. What about lymphocyte counts? The latter is particularly relevant given the demonstration that a high neutrophil to lymphocyte ratio correlates with venous thrombosis risk in PV.

Response to Reviewer 3 comment:  We thank the reviewer for his/her question. As the same reviewer affirms, the neutrophil-to-lymphocyte ratio is an established thrombosis risk factor for PV (Carobbio A, Vannucchi AM, De Stefano V, Masciulli A, Guglielmelli P, Loscocco GG, Ramundo F, Rossi E, Kanthi Y, Tefferi A, Barbui T. Neutrophil-to-lymphocyte ratio is a novel predictor of venous thrombosis in polycythemia vera. Blood Cancer J. 2022 Feb 10;12(2):28. doi: 10.1038/s41408-022-00625-5. PMID: 35145055; PMCID: PMC8831521.) and it also impacts prognosis of MPN patients (Larsen MK, Skov V, Kjær L, Eickhardt-Dalbøge CS, Knudsen TA, Kristiansen MH, Sørensen AL, Wienecke T, Andersen M, Ottesen JT, Gudmand-Høyer J, Snyder JA, Andersen MP, Torp-Pedersen C, Poulsen HE, Stiehl T, Hasselbalch HC, Ellervik C. Neutrophil-to-lymphocyte ratio and all-cause mortality with and without myeloproliferative neoplasms-a Danish longitudinal study. Blood Cancer J. 2024 Feb 9;14(1):28. doi: 10.1038/s41408-024-00994-z. PMID: 38331919; PMCID: PMC10853217.). For this reason, we decided not to repeat the evaluation.

We add this limit and the two quotes to the discussion section.

Reviewer 3 comment:

  1. Although the article is not written by native speakers of English, it is, to the authors’ credit, generally easy to follow. However, it does need to be reviewed by a native speaker of English or someone highly proficient in the language. For example, the statement about “women who respect men” makes no sense and needs to be reworded.

Response to Reviewer 3 comment:  We thank the reviewer for his/her correction. In the revised version, we perform an English revision of the language.

Reviewer 2 Report

Comments and Suggestions for Authors

I have the following specific suggestions/comments:

1. Not sure why the title shows strange text, and why not corrected by authors before submission: MI-Croparticles, Nega-Tive

2. The current recommended abbreviation for vWF is 'VWF; please change throughout. The recommended abbreviation for FVIII C is 'FVIII:C'; please change throughout.

3. Introduction; entire page 2 and page 3 contains many paragraphs, and statements of fact, but hardly any references; please include a reference for each statement of fact. Occasionally, authors state 'Several studies report.." and then list only a single reference. Many times they state 'In literature,' or 'Literature suggests', but fail to provide a reference.

4. Page 4: "The study protocol was by the declaration of Helsinki." ?

5. Page 4: "centrifuging samples at 2000×g for 20 min at room temperature, then divided in small aliquots and stored at −80 °C until testing." Including for microparticles? Also for lupus anticoagulant?

6.  "ADAMTS-13 activity was assayed using collagen-binding enzyme immunoassay." may be misunderstood, since these assays look for the ability of plasma to cleave VWF, with VWF:CB as a detection system. Does Ref 13 as cited really describe the assay they used?

7. "According to the literature, a commercial chromogenic substrate assay measured FVIII C [14]." Does ref 14 as cited really describe the assay they used? Most labs use a one stage assay, not a chromogenic assay. Did you authors actually measure FVIII:C, or get their local hemostasis lab to measure FVIII:C?

8. "A correction factor on FVIII and vWF was applied according to the AB0 group and aging as reported in the literature" ? what correction factor? not mentioned in results.

9. Table 1: Not labelled as such, and no footer describing table abbreviations. Group D FVIII:C listed as 11.47±8.3; authors may mean 114.7±8.3? Check other data entries for errors.

10. Ref list: Ref 28 and 29 are one ref split into 2; authors should check/correct in text citations from ref 28.

Comments on the Quality of English Language

Most of the English language is fine; however, there are occasional issues, including as flagged above

Author Response

(The authors gave the same response as above.)

Reviewer 3 Report

Comments and Suggestions for Authors

- Please pay attention to typos in the title

- Page 4, line 149: please provide more evidence concerning absence of acute inflammation.

- Introduction: as this manuscript is more about diagnosis and markers of a pathological condition, talking about symptoms in three paragraphs (page 2, paragraph 2,3,4) is not suitable.

- The same issue with treatment of this kind of patients in almost all page 3. Instead, they could provide more information about neutrophil and monocyte roles in this context.

- Page 9, line 252: please rephrase your sentence.

- Page 9, line 275-278: please rephrase your sentence.

- Some proteins or antibodies interfere with the tests the authors had been performed. Do the authors have information about fibrinogen concentration, rheuma factor, and lupus anticoagulant for these patients?

Comments on the Quality of English Language

I have suggested some improvements.

Author Response

(The authors gave the same response as above.)

Round 2

Reviewer 2 Report

Comments and Suggestions for Authors

There remain some issues:

1. ADAMTS13 in title to match the main text

2. I still don't understand their use of a supposed correction factor for O-blood group; also: "A correction factor on FVIII and VWF was applied according to the AB0 group (blood group 0 levels were increased by 30%) as reported in the literature" is incorrect; FVIII and VWF levels are some 20-30% lower (not higher) in O-blood group.

3. Reference 25 citation needs fixing.

Comments on the Quality of English Language

mostly fine

Author Response

Dear Editor,

Enclosed is our revised version of the manuscript entitled “ADAMTS13, von Willebrand Factor, Platelet MIcroparticles, Factor VIII, and Impact of Somatic Mutations in the Pathogenesis of Splanchnic Vein Thrombosis Associated with Bcr-Abl Negative Myeloproliferative Neoplasms” We want to thank the Editor and the reviewers for their constructive criticism, which substantially improved our paper. We hope that the paper will be suitable for publication. We have made a point-by-point answer to the referees’ comments below.

Answers to Reviewer 1

Reviewer 1 comment: There remain some issues:

  1. ADAMTS13 in title to match the main text

Response to Reviewer 1 comment:  We thank the reviewer for his/her correction. In the revised version, we corrected the title. Moreover, we checked the most used abbreviation on Pubmed, and we can confirm that the version suggested by the reviewer (ADAMTS13) is the most used on Pubmed compared to (ADAMTS-13 or ADAMTS 13).

Reviewer 1 comment: 2. I still don't understand their use of a supposed correction factor for the O-blood group; also: "A correction factor on FVIII and VWF was applied according to the AB0 group (blood group 0 levels were increased by 30%) as reported in the literature" is incorrect; FVIII and VWF levels are some 20-30% lower (not higher) in O-blood group.

Response to Reviewer 1 comment:  We thank the reviewer for his/her question. We fully agree with the reviewer; our corrections were made according to the literature. To explain our correction, we can use two examples:

  • Without correction, A group measure is 100, and 0 group is 70 (30% lower than other groups). With a 30% increase correction, A group is always 100, and 0 groups is 70 + 21 (30% increase) = 91
  • Without correction, A group measure is 100, and 0 group is 80 (20% lower than other groups). With a 30% increase correction, A group is always 100, and 0 groups is 80 + 24 (30% increase) = 104

To improve the test, we changed the sentence: “Normally, blood 0-group levels are some 20-30% lower than those of non-0-groups. To correct this bias, 0-group levels were increased by 30%”

Reviewer 1 comment: 3. Reference 25 citation needs fixing.

Response to Reviewer 1 comment: We thank the reviewer for his/her correction. In the revised version, we correct reference 25.

Answers to Reviewer 2

Reviewer 2 comment: - The authors have improved their manuscript.

Response to Reviewer 3 comment:  We thank the reviewer for his/her positive comment.

Reviewer 2 comment: Please change PCR to CRP on page 4.

Response to Reviewer 2 comment:  We thank the reviewer for his/her correction. In the revised version, we change PCR to CRP on page 4.

Reviewer 3 Report

Comments and Suggestions for Authors

The authors have imrpoved their manuscript. Please change PCR to CRP on page 4.

Author Response

(The authors gave the same response as above.)
